# Multisensory stimuli facilitate low-level perceptual learning on a difficult global motion task in virtual reality

Catherine A. Fromm[1], Ross K. Maddox[2], Melissa J. Polonenko[2], Krystel R. Huxlin[3]*, Gabriel J. Diaz[1]

1 Chester F. Carlson Center for Imaging Science, Rochester Institute of Technology, Rochester, New York, USA, 2 Department of Brain and Cognitive Science, University of Rochester, Rochester, New York, USA, 3 Flaum Eye Institute, University of Rochester, Rochester, New York, USA

* khuxlin@UR.Rochester.edu

**Data availability statement:** All data files are available from the Open Science Framework database (https://osf.io/6qmu2).

## Abstract

The present study investigates the feasibility of inducing visual perceptual learning on a peripheral, global direction discrimination and integration task in virtual reality, and tests whether audio-visual multisensory training induces faster or greater visual learning than unisensory visual training. Seventeen participants completed a 10-day training experiment wherein they repeatedly performed a 4-alternative, combined visual global-motion and direction discrimination task at 10° azimuth/elevation in a virtual environment. A visual-only group of 8 participants was trained using a unimodal visual stimulus. An audio-visual group of 9 participants underwent training whereby the visual stimulus was always paired with a pulsed, white-noise auditory cue that simulated auditory motion in a direction consistent with the horizontal component of the visual motion stimulus. Our results reveal that, for both groups, learning occurred and transferred to untrained locations. For the AV group, there was an additional performance benefit to training from the AV cue to horizontal motion. This benefit extended into the unisensory post-test, where the auditory cue was removed. However, this benefit did not generalize spatially to previously untrained areas. This spatial specificity suggests that AV learning may have occurred at a lower level in the visual pathways, compared to visual-only learning.

## Introduction

As an individual grows, their visual system must develop in response to ever-changing demands. These changes can be short-term adaptations, which quickly fade when conditions change, or can take the form of longer-lasting changes in response to *perceptual learning* [1,2]. Laboratory-based investigations of perceptual learning have found that long-term changes in an observer's ability to detect, discriminate and identify sensory stimuli can also be brought about through multiple forms of training (e.g., [3–5]). Visual perceptual learning has also been explored for therapeutic applications. For instance, it has been used to partially restore visual perceptual abilities after stroke (e.g., [6–12]).

**Funding:** Research reported in this publication was supported by a grant from the National Eye Institute of the National Institutes of Health under award number R15 EY031090, and by an Unrestricted Grant from Research to Prevent Blindness to the University of Rochester's department of Ophthalmology.

**Competing interests:** The authors have declared that no competing interests exist

In the laboratory, visual perceptual learning is commonly induced through the repeated exposure to a psychophysical task that tests sensitivity to and/or manipulates a property of a visual stimulus, such as contrast and spatial frequency [13], orientation [14], or motion direction [15]. This stimulus property may be systematically adjusted over time with the goal of maintaining near-threshold performance, even as sensory thresholds are reduced and sensitivity is increased in response to training. Most paradigms involve multiple training sessions that are completed on consecutive days and are flanked by pre and post tests. These pre/post tests provide insight into the overall magnitude of improvement, and an opportunity to introduce novel manipulations meant to explore the limitations of perceptual learning.

Investigations into perceptual learning have revealed that, in certain training contexts, learning is restricted to the trained task [16], or retinotopic location [17]. This might arise from a reweighting of neuronal activity at the earliest levels of processing, where there is specificity for certain stimulus features and spatial locations [18–20]. However, other mechanistic hypotheses for perceptual learning have also been put forth (see reviews [21,22]), including changes in neuronal tuning [20], changes in population responses [23] or changes in readout of low-level sensory activity by higher-level areas [24]. There has been some success mitigating the spatial specificity of perceptual learning through the use of alternative training paradigms. These methods include double training, where a separate task is trained simultaneously in a second location, [25], brief pretests at locations that are not revisited during training [26], increasing the duration of training [27], stimulus variability [28], the manipulation of visual attention [29–33] and other strategies involving spatial variability. Although mechanisms of spatial transfer are not yet fully defined, and may vary depending on training paradigms employed, they are generally thought to involve a shift in processing away from early visual areas, towards higher levels of the processing hierarchy [34–36].

There has also been some success in increasing the speed of training through the use of multimodal stimuli [37]. For instance, audiovisual training has been shown to induce greater and faster visual improvements - both within sessions and across sessions/days - suggesting that adding an auditory component may influence the locus of neural plasticity by invoking multisensory integration [38–40].

The present study examines the potential benefits of multisensory integration for learning in healthy individuals using a visual/auditory task presented within a virtual reality (VR) training environment. There are several reasons why virtual reality is a potentially impactful method to deploy multisensory training paradigms. Considerable technological advances in the rendering of spatial audio make it possible for the task-paradigm to leverage stationary or dynamic auditory sources that listeners are able to localize within the 3D environment. As was recently demonstrated by Alwashmi et al. [41], this improved spatial rendering can facilitate audiovisual binding, which has been shown to be an important element of visual/auditory perceptual learning [42]. Additionally, the use of VR may alleviate the prohibitive requirement that spatially-localized training must occur in the lab, where the use of an eye tracker allows the experimenter to detect where the participant is fixating while a stimulus is delivered to a particular location in the visual field. A subset of modern VR headsets now include integrated eye tracking intended for widespread use, that is intuitive to employ, and that can be operated without training.

This study asks if audiovisual training can enhance learning of visual motion direction discrimination, with both visual and auditory stimuli created and presented in a gaze-contingent

manner, within a VR environment, as depicted in Fig 1. Task difficulty was controlled using a staircase procedure that manipulated the direction-range of the coherent portion of dots included in the random dot stimulus (Fig 2). We asked whether learning occurred in VR, whether it occurred at a faster rate in a group presented with a visual-only training stimulus (the V group) or when the visual stimulus was accompanied by a spatialized auditory component (the AV group). We also asked if learning demonstrated by the AV group was retained when the auditory cues were removed, and if learning transferred to untrained visual field locations. Finally, by making the auditory stimulus informative only about the movement of the motion stimulus along the horizontal axis, we were able to test whether the benefits of training were direction specific - a feature which is indicative of learning occurring at an earlier stage of visual processing [34,37,38].

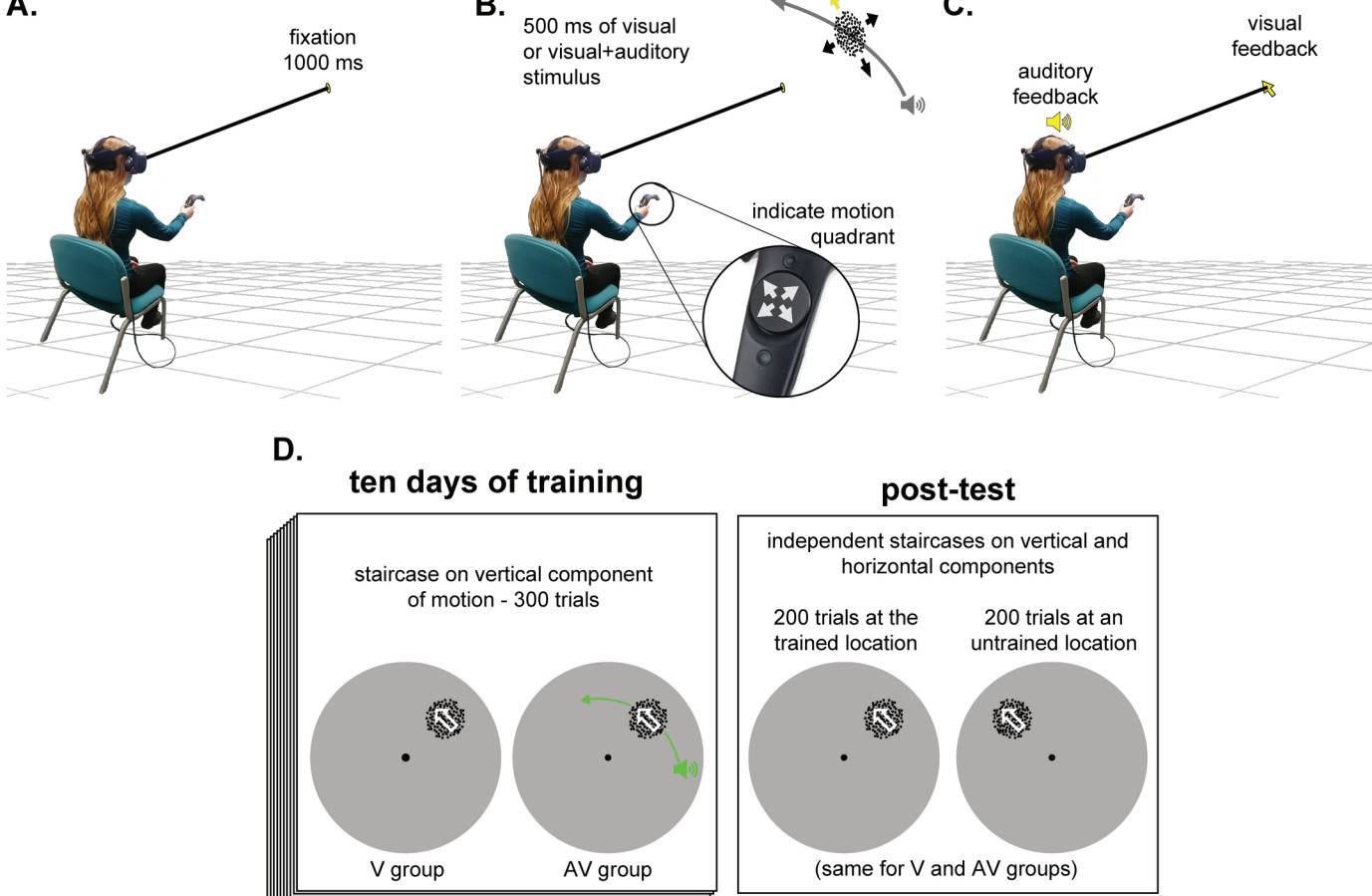

**Fig 1. The Experimental Task.** (A) Participants were seated and wearing the HTC Vive Pro Eye virtual reality display. Fixation and head orientation were enforced with the integrated eye and head trackers. (B) The visual stimulus consisted of dots moving within a 5° in diameter aperture located at 10 ° azimuth and elevation from fixation, and 0.57 m in depth. The global motion of the visual stimulus was aligned with one of four quadrants. During training for the AV group, the visual stimulus was accompanied by a suprathreshold, spatialized, auditory stimulus that arced horizontally through the visual stimulus in a direction consistent with the horizontal component of its global motion (as depicted). The participant indicated their perceived global motion direction for the visual stimulus using the thumb-pad on the HTC Vive control wand. (C) The participant's response was followed by visual feedback about the true direction of global motion and auditory feedback about the correctness of their response. (D) Each group completed 300 trials per day for 10 days of training. The post-test involved 200 trials performed in the absence of an auditory stimulus at both the trained and an untrained visual field location (400 trials total).

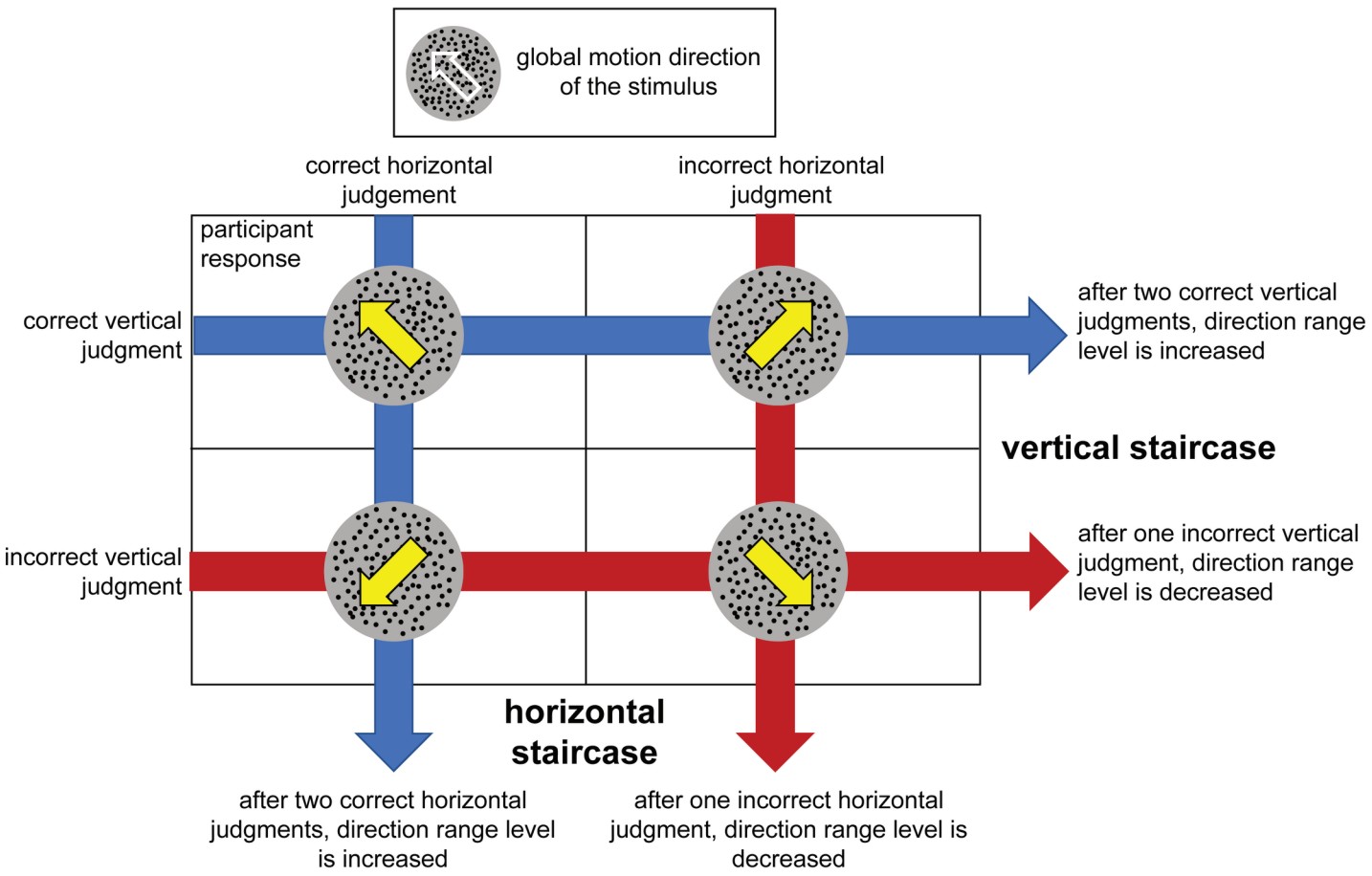

**Fig 2. Trial outcomes for each possible participant response.** Task difficulty was controlled using a staircase procedure that manipulated the direction-range of the coherent portion of dots included in the random dot stimulus. During training, because the AV group was provided suprathreshold auditory information about the horizontal component of stimulus motion, the staircase was on the vertical component of judgments (for both groups). However the post-test that was conducted in the absence of an auditory stimulus involved two independent staircases with interleaved trials - one that was modulated by the correctness of judgments of the vertical component of motion, and one by the horizontal. The outcomes depicted in this example assume a stimulus with a global motion direction towards the northwest quadrant.

## Materials and methods

### Participants

Participants were recruited from the Rochester Institute of Technology (RIT) campus community between the dates of 2020-10-05 and 2021-08-03. The experiment was approved by the Institutional Review Board at RIT. Prior to participation, while under supervision of the experimenter, individuals were provided details of the experimental procedure and aims, and all chose to sign the associated informed consent forms. Participants had normal or corrected-to-normal vision, and self-reported as having no documented hearing or auditory processing disorders. A total of 17 participants took part in the full 12 days of the study, 6 who self-identified as female and 11 who self-identified as male. Eight participants were assigned to the V group, and 9 to the AV group. The mean age (±SD) was 25.2 ± 3.2 years. Six participants (3 female) attended the first session (practice block) only. They either opted out

of the study due to discomfort using the virtual reality system or they were excluded by the experimenter because of unusually poor task performance in the practice block.

## Apparatus

During all experimental sessions, participants were seated, wearing an HTC Vive Pro Eye virtual reality headset (HTC Corporation, Taoyuan City, Taiwan). The binocular field of view of the headset is nominally 110° horizontally, but this can vary within a few degrees with the distance of the eyes from the headset's optics. During pilot testing, we verified that the task environment was able to maintain the nominal update rate of approximately 90 Hz. The experimental software was created and rendered using Unity version 2019.1.14f1, and run on a PC equipped with an Intel i7-6700 CPU and an NVIDIA 2080 RTX graphics card. The eye tracker integrated into the HTC Vive Pro Eye was used during each session, and controlled using the manufacturer-provided SRAnipal plugin to Unity, version 1.3.2.0.

At the start of each testing or training session, participants adjusted the headset to improve comfort and display quality, before completing the integrated Vive Pro Eye calibration routine. Following calibration, a custom calibration assessment was run through Unity. Participants were presented with a visual spherical target sequentially presented at one of 9 locations arranged in a grid pattern, ±15° wide. Targets were yoked to the head position. All targets were at a virtual depth of 0.57m, which was experimentally determined to be the virtual image plane of the optics in the specific headset used for this experiment. This depth was selected to minimize the effects of the vergence accommodation conflict, which has been known to cause discomfort during use of virtual reality displays [43]. Data were recorded for a period of 500ms during fixation of each target, and error was measured between the true target direction and the gaze direction reported by the eye tracker. The average error across all sessions and all participants at the central fixation point was 1.1° with a standard deviation of 0.4° of visual angle.

Once the calibration assessment concluded, the seated participant was instructed to adopt a comfortable head pose that could be maintained for a long period of time. While in this pose, the participant's head position in the 3D world reference frame was recorded so that it could subsequently be used to correct head pose over the course of the experiment, similar in spirit to the use of a fixation point to constrain gaze direction. At the start of each trial, head pose was compared to the recorded reference pose, and if it was found to have rotated from the reference orientation by more than 2°, a realignment procedure began. To realign the head, participants moved their head to align a rectangular box that was stationary in head-centered reference frame with a bar that was stationary at the world reference frame. This world-fixed box was located at the fixation point, parallel to the ground, and perpendicular to the vector between the point at the center of the head and the fixation point. Trials would not begin unless fixation was within 0.3° of the fixation point, and head pose was within 2° of the world-fixed box for one uninterrupted second. The trial was aborted if gaze deviated by more than 1.5° from fixation mid-trial, if head orientation deviated by more than 2° from the designated pose, or if position deviated by more than 10 inches from the position defined by the resting pose. When a trial was aborted, the staircase procedure reset to the conditions at the end of the previous trial.

## Motion discrimination task

Whether during the initial orientation, titration and practice session, the training sessions that ensued, or the post-test, participants were asked to discriminate the global direction of

motion of a field of randomly distributed dots moving within a 5° diameter aperture, as represented in Fig 1. Global motion was in one of four oblique directions (45° from horizontal): upper right, lower right, upper left, or lower left. As each session progressed, the individual dot motion directions were perturbed away from the 45° oblique direction (as described in the section ) thus increasing the range of dot directions over 10 discrete levels (0, 40, 80, 120, 160, 200, 240, 280, 320, 355°). The direction-range level changed from trial-to-trial according to a 2-up-1-down staircase with a convergence level of 70.7%.

## Generation of visual stimuli in VR

We used random dot stimuli presented within a 5° diameter aperture. Individual dots were 14 arcmin in diameter and dot density was 3.5 dots/$°^2$. Dots moved at 10°/s, and had a lifetime of 250ms before disappearing and respawning in a new, random location. The total duration of the stimulus was 500 ms. Difficulty of the visual task was parameterized in two ways. First, we manipulated the proportion of "noise" dots, which moved with random trajectories as opposed to the "signal" dots, which moved with coherent trajectories. The procedure for setting this proportion is described in detail in the section . Second, we varied the range of directions in which signal dots could move. This was dynamically changed within a session, using a staircase procedure tied to performance, as described earlier and represented in Fig 2.

## Generation of auditory stimuli in VR

During training, the AV group was presented with both the visual stimulus and a concurrent, spatialized, auditory stimulus. The spatialized auditory stimuli implemented with the SteamAudio spatializer plugin for Unity3D simulated the noise being produced from the location of an emitter in the simulated 3D environment. We chose to restrict the motion direction of the auditory stimulus to the horizontal direction because, whereas judgments of horizontal source location rely on binaural differences in sound arrival time and sound level, judgments of vertical motion direction rely primarily on changes in spectral characteristics, and it is easier to simulate and perceive horizontal than vertical sound motion [44]. Because oblique directions would have required a combination of the two sound directions, they were not considered optimal for the present experimental design. The parameters of the auditory stimulus were specifically chosen to make the horizontal movement direction of the AV group's auditory stimulus well above threshold. The invisible auditory emitter's initial position was 20° left/right of the visual stimulus center, and it moved at 80°/s along an arc with radius 0.57m that was parallel to the ground plane (i.e., horizontal) and centered on the observer's head. The emitter became silent when it reached a point 20° on the other side of the visual stimulus 500 ms after onset (Fig 1B). The sound emitted was pulsed white noise with a pulse frequency of 12Hz. Sound location was simulated using a generic head-related transfer function (HRTF). The decision to restrict the movement of the auditory stimulus to the horizontal direction rather than vertical was motivated, in part, by the finding that there is potential for up/down confusion when using generic HRTFs in virtual spatialized audio [45,46] but more clarity in left/right discrimination. Moreover, the benefits of using them are primarily seen when making absolute spatial localization judgments [46], and not judgments of motion direction. Pilot testing suggested that hearing individuals would have no trouble accurately and consistently judging the motion direction of the auditory source.

## Experimental procedures

The first day of the experiment involved a procedure designed to ascertain the percentage of random noise dots needed in the visual stimulus to ensure that all participants started with similar direction discrimination and integration performance at the outset of the study. Auditory stimuli were not presented during this initial session. Participants first completed 20 practice trials with initial difficulty set at 30% of the dots moving randomly. This allowed familiarization with the apparatus and task (see the section  for further details on random dot settings). If the practice block was so difficult that the participants could not progress, or so easy that every trial was correct, the proportion of randomly-moving dots was adjusted in increments of 10% noise dots until the participant was able to maintain a direction range threshold of at least 80° within the first 20 trials. Following this coarse adjustment of the baseline signal:noise dot ratio, participants performed a titration block of 100 trials. Performance in this block was used to fine-tune the signal:noise dot ratio for the rest of the experiment. After the titration block, participants took a brief break out of the headset while the experimenter fit a psychometric function to their trial-by-trial results (see the section  for further details on the fitting process). If the direction range threshold of the psychometric function was between 100° and 200° , this signal:noise ratio setting was kept for the remainder of the study. If direction range threshold performance was above 200° or below 100, difficulty was adjusted by adding or subtracting 5% noise dots and the experiment proceeded. The final percentages of noise dots was 31.5% for the V group (SD: 12.6%) and 35.5% for the AV group (SD: 17%), with a two-tailed independent-samples t-test indicating no difference between them (t(15) = 0.551, $p$ = 0.59).

Participants were divided into two groups that determined the nature of the stimuli employed for training: the *visual* training group, and the *audiovisual* training group, hereafter referred to as the V and AV groups. There were no auditory stimuli presented during post-tests, or during training for the V group. However, during training trials for the AV group, a spatialized auditory stimulus was always presented simultaneously with each visual stimulus. Each training session consisted of a single block of 300 trials of the 4AFC direction discrimination task. At the start of each trial, a fixation point was presented in the center of the participant's visual field at a virtual depth of 0.57 m. After the eye tracker recorded 1s of fixation on the target, and the head tracker simultaneously recorded 1 s of stationary head in the home position, the trial proceeded to the stimulus. If the eyes or head deviated during this 1 s interval, the trial reset. Additionally, if the eyes or head deviated from their positions at any time during the stimulus presentation, or before the participant provided a response, the trial reset. After the 1 s interval ended, during continued fixation, a visual or visual+auditory motion stimulus was presented for 500 ms at the same depth as the fixation target, 10° azimuth and 10° elevation away from fixation (trained location, Fig 2A). The participant was able to indicate perceived motion direction (upper right, lower right, upper left, lower left) at any point following stimulus presentation by selecting one of 4 corner regions on the HTC Vive controller's trackpad. There was no limit placed on the response time. After responding, participants received feedback in the form of an arrow presented at fixation, illustrating the correct direction of the motion (Fig 1C). They also received positive or a negative auditory feedback denoting correctness of the response. This feedback was delivered in the form of sounds similar to those used in video games, and easily discerned.

Participants were brought back to the lab 1 day after the last day of training for assessment of performance on a visual version of the 4AFC direction discrimination task at two locations: the trained location at 10° azimuth and elevation from fixation, and at an untrained location centered at -10° azimuth and 10° elevation. The task involved using two interleaved

staircases: one tracked correctness of the judgment of horizontal component of the visual motion stimulus while the other tracked correctness of the vertical component judgment. No auditory stimuli were presented. When participants scored 2 correct trials in a row at a particular direction range level on either the horizontal or vertical components, depending on which staircase that trial corresponded to, direction range in the stimulus was increased by 40°, making the task harder. When they scored an incorrect response, direction range was decreased by 40°, making the task easier.

## Analysis

To assess performance in each session, data was exported from Unity as a custom CSV file, which was then imported into Matlab version R2018a for quantitative analysis. Psychometric functions were fit to each session using custom code to calculate the session's direction range threshold (DRT), a measure of direction integration ability. DRTs were calculated by fitting a Weibull function to the percentage of correct judgments made at each level of direction range.

Weibull function parameters were fit using maximum likelihood estimation implemented in MATLAB using the Optimization Toolbox. The chance level for each component judgment was 50% and the fits were performed assuming a 5% lapse rate. The Weibull function was evaluated to find the direction range level at which participants answered correctly 72.5% of the time, the value halfway between ideal performance considering the lapse rate and chance.

Although participants made a judgment on both vertical and horizontal components of the global motion stimulus simultaneously when selecting one of four oblique motion directions on each trial, separate thresholds were calculated for judgments of the vertical component direction and judgments of the horizontal component of motion for each training session. For example, if the correct direction was to the upper left, and the participant selected the upper right, the participant would score correctly on the judgment of the vertical component, but incorrectly on the judgment of the horizontal component (see Fig 2). Because the vertical motion direction of the stimulus was specified in a manner that was independent from the horizontal component of stimulus motion, having perfect information about the stimulus' movement along one of those axes provided no meaningful information about the direction of motion along the orthogonal. Without knowledge of possible statistical dependencies between the orthogonal motion components of judgments, it made sense to analyze these dimensions separately. As such, analysis focused on the judgment of combined motion direction, which combined vertical and horizontal components, and on the vertical component of motion. Analysis did not focus on the horizontal component of motion because the AV group was provided with a supra-threshold, dynamic auditory cue, corresponding reliably to the horizontal component of the visual stimulus' global motion direction on every trial.

Vertical and horizontal training rates were calculated as the slope of a linear fit to the mean DRT across the 10 training sessions. The decision to analyze vertical motion direction threshold separately from the horizontal threshold made comparison of training efficacy between the unisensory and multisensory groups possible. Learning rate in the vertical component was comparable between groups as it was a visual discriminant for both the V and AV tasks. For the horizontal component, no comparison could be made between the training rate in the two groups because the performance rapidly reached ceiling levels for this component in the AV group, who received unambiguous, suprathreshold information about the horizontal component of motion direction from the purely horizontal auditory motion stimulus.

**Statistics.** Linear mixed-effects models were fit to the performance data and post-hoc tests were conducted to determine the impact of participant (as a random effect), and fixed effects including some combination of group (V or AV), training or post-test session, motion

component (global, vertical, horizontal), and training location (for the post-test session, whether in the trained or untrained location) on DRT thresholds. These models were fit using the lme4 package [47] in the R statistical software environment (version 4.0.5 [48]). Degrees of freedom were approximated using the Kenward-Roger method [49], and the effect size ($\eta_p^2$) was calculated using the effectsize package version 0.8.3 [50]. Post-hoc tests were performed using the emmeans package [51], and its function *eff_size* for the calculation of Cohen's *d*.

## Results

### Learning occurred at similar rates for the V and AV groups

Analysis of training days 1–10 reveals clear signs of perceptual learning for DRTs in both training groups (Fig 3). Statistical investigation involved the application of separate linear mixed effects models to the DRT calculated from judgments of the global and vertical components of motion. In each case, the participant's study ID was included as a random factor, and fixed effects included the factor of group assignment (V/AV) and day (1–10) as a continuous numeric variable.

For the model fit to the DRT calculated from judgments of the combined direction of motion (e.g., accounting for both the horizontal and vertical component of motion; Fig 3A), the random factor of patient ID accounted for 64.7% of overall variance. There were significant effects of group (F(21.164, 151) = 2.97, *p* = 0.007, $\eta_p^2$ = 0.34) and day (F(21.164, 151) =

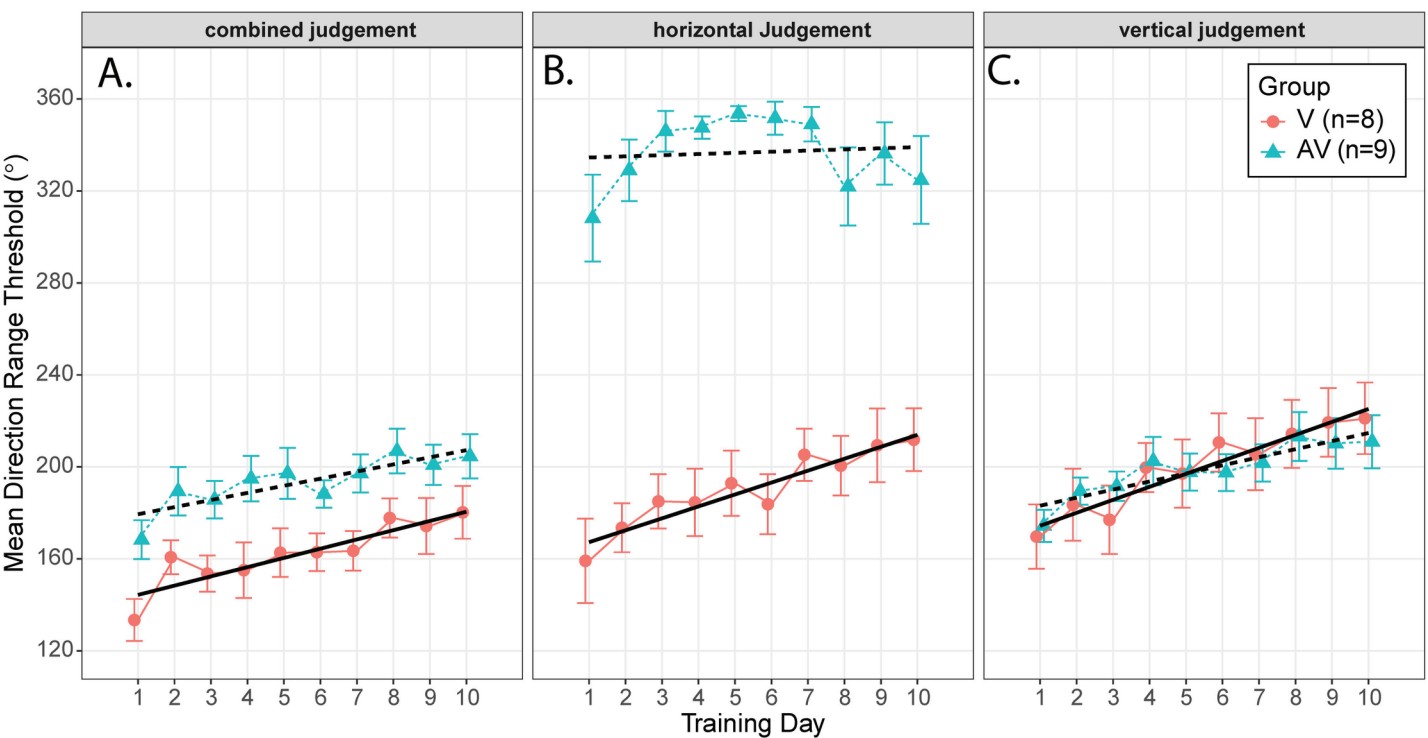

**Fig 3. Training results:** Direction range thresholds (DRT) for each of the 10 days of training broken down by group and motion component. (A) Combined judgments are considered correct only when the participants response correctly identifies both the horizontal and vertical motion components of the visual stimulus correctly (B) DRT for the horizontal component of motion. Horizontal judgments for the AV group (triangle symbols, center panel) are at ceiling due to the presence of a suprathreshold dynamic auditory cue that was introduced at the same time as the visual stimulus, and that moved through space in the same direction of the horizontal component of the visual stimulus. Error bars reflect the standard error of the mean. (C) DRT for the vertical component of motion.

5.123, $p < 0.001$, $\eta_p^2 = 0.3$), indicating that both groups improved over the course of learning, despite differences in overall performance. Learning rates are indicated by the marginal slope estimates, which for the AV-group was 3.09°/day with standard error of 0.604, and for the V group, the slope estimate was 4.01°/day with standard error of 0.641. The lack of a significant interaction between group and day (F(21.164, 151) = −1.049, $p = 0.29$) fails to support the hypothesis that training with the AV stimulus improves overall performance on combined judgments.

There was a slight difference of learning rates apparent in judgments concerning the vertical component of motion that is apparent in Fig 3C. A model fit to DRT for the vertical component of motion included fixed effects of group assignment and day and a random factor of patient ID that accounted for 76.6% of overall variance. The effect of day was significant (F(18.408, 151) = 8.809, $p < 0.001$, $\eta_p^2 = 0.41$), indicating that performance increased overall during training. Although the interaction between day and group was also significant (F(18.408, 151) = 8.809, $p = 0.017$), it had a very small effect size ($\eta_p^2 = 0.04$), meaning that the groups had significantly different learning rates for judgments of the vertical component of motion. The learning rate indicated by the marginal slope estimate for the AV group was 3.513°/day with standard error of 0.60°/day, and for the V group, the slope estimate was 5.646°/day with standard error of 0.64°/day. Thus, the AV group learned more slowly than the V group for the vertical component of motion.

The learning rate was not investigated for judgments of the horizontal component of motion presented in Fig 3B because the AV group had suprathreshold information about horizontal motion direction during training while the V group did not. The observation that the AV group's DRT was near the ceiling of 360 degrees suggests that the auditory cue to horizontal motion direction was driving behavior for the AV group.

In summary, both V and AV training groups exhibited visual motion discrimination and integration learning in their peripheral visual field, and there was a slightly higher rate of learning for the V than the AV group for the vertical component of motion.

## Performance levels on the final day of training were maintained during the visual-only post-test at trained locations

A central question of this study was whether the improvement in performance facilitated by training with an auditory accompaniment to the visual stimulus facilitated unisensory visual judgments of motion direction. Fig 4 presents a visual comparison of performance on the final day of training (i.e., t10 in Fig 4; green bars) with post-test performance at the trained location (orange bars) for each combination of motion component and group. This visual comparison suggests that performance levels were almost uniformly maintained between the last day of training and the post-test. Because training never occurred at the untrained locations assessed in the post-test, a linear mixed model was used to test thresholds measured only at the trained locations (orange and green bars in Fig 4). Fixed effects included day (t10 and post-test), group (V and AV), and motion component (vertical and horizontal). The random effect of participant ID contributed to 10% of the overall variance in the model. The application of consecutive contrasts found a difference between the AV group's judgments of horizontal motion direction (T(44.4) = 3.015, $p = 0.004$, $d = -1.60$), indicative of the slight drop visible in Fig 4. In summary, all groups demonstrated persistence of learning from t10 to the post-test at trained locations, albeit at a slightly reduced level for AV group's judgments of horizontal motion direction, which is not surprising given that the suprathreshold auditory cue present for t10 was no longer available for the post-test.

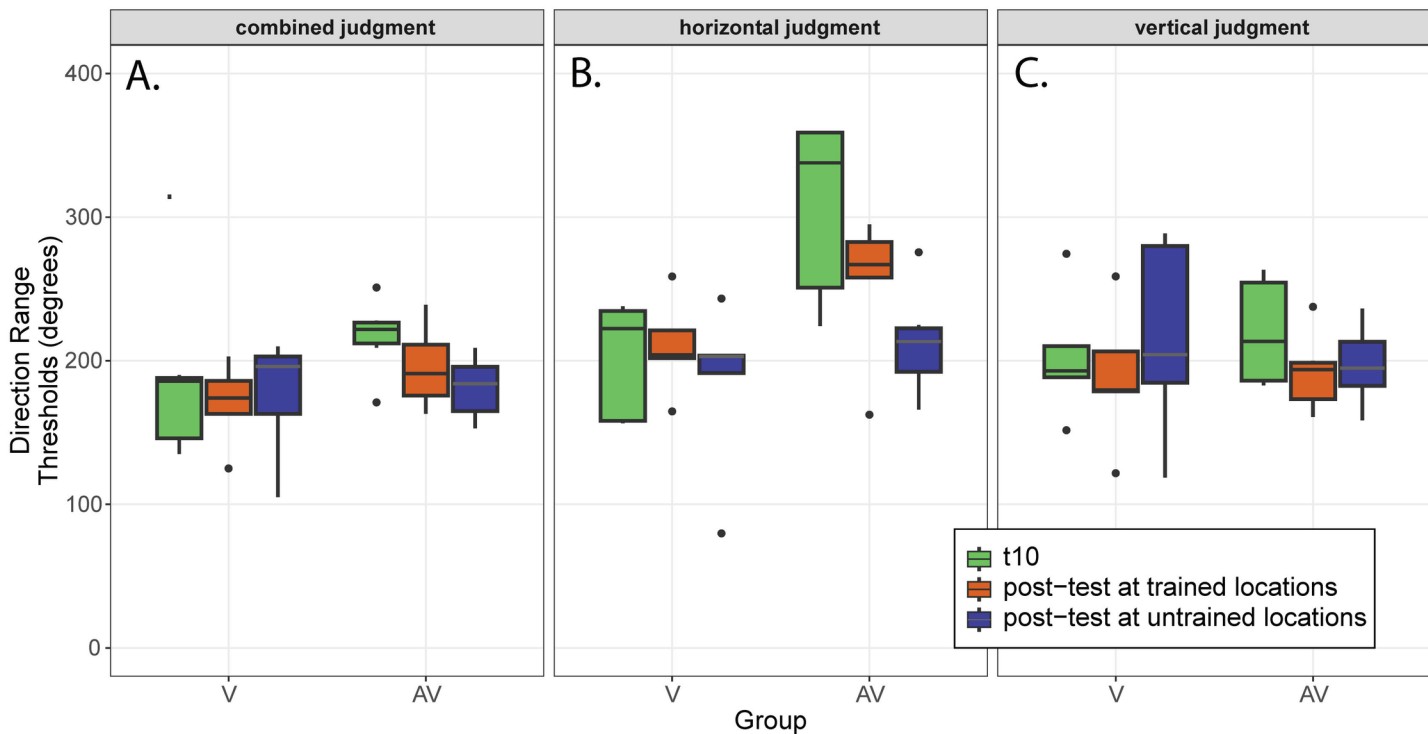

**Fig 4. Direction range thresholds (DRT) on the final training day (t10) and post-tests at both trained/untrained locations**. Thresholds are broken down by motion component and participant training group (V or AV). Higher DRT are indicative of better performance. Lower and upper hinges of the boxplot represent the 25th and 75th percentiles around the mean (black solid line). Whiskers extend to the largest value no further than 1.5 inter-quartile ranges from the hinge. (A) DRT for the global motion judgment. (B) DRT for the horizontal component of motion. Data for the AV group suggests the more accurate perceptual judgments that were present during training in the presence of an auditory cue to horizontal motion direction persisted from t10 (AV group, green bar) to the post-test (AV group, orange bar), despite the absence of the auditory cue. Results also suggest that this effect did not transfer spatially to untrained locations (AV group, blue bar). (C) DRT for the vertical component of motion.

### Training with an auditory cue to horizontal motion specifically benefited post-test visual judgments of horizontal, but not vertical motion direction, at trained locations

Visual inspection of Fig 4B suggests that the AV group's high levels of performance for horizontal judgments during the post-test did not extend to post-test judgments concerning the vertical direction of motion. Contrasts applied to each combination of group and day revealed a difference in performance between both horizontal and vertical motion directions on both day t10 ($T(43.1) = -6.103$, $p < 0.001$, $d = -2.635$) and on the post-test ($T(43.1) = -2.834$, $p = 0.007$, $d = -1.471$), providing further support for the interpretation that the advantage held by the AV group during training for judgments of horizontal motion direction persisted into the post-test despite the removal of the auditory cue that was present during training. However, the benefits of training with a horizontal auditory cue did not appear to extend to judgments of the vertical motion component when this cue was removed in the post-test.

### The benefit from AV training was retinotopically specific

For all conditions, post-hoc judgments of motion direction at untrained locations demonstrated a benefit from training. However, there was no evidence that the AV group *outperformed* the V group at untrained locations. This is apparent through visual inspection

of Fig 4B, which shows that although the AV ability to judge the horizontal component of motion at trained locations (orange bar) did not fully transfer to untrained locations (blue bar), there was partial transfer. This partial transfer resulted in performance that was on-par with judgments of vertical motion direction, which did not benefit from the presence of the auditory stimulus during training (Fig 4C, AV group, blue bar). This interpretation is supported by the application of a linear mixed effects model to test differences between t10 and the post-test at untrained locations, with fixed effects of day, group, and motion component, and a random effect of participant ID. Consecutive contrasts applied between t10 and post-test sessions for each group and component reveal that the only change in performance between the last day of training and the post-hoc test at untrained locations was for the AV group on the horizontal component of motion (T(45) = 5.34, $p < 0.001$, $d = -2.5$), indicative of the decrease in performance when tested at untrained retinal locations. Moreover, pairwise contrasts of motion component applied within each combination of group and day failed to find a significant difference in performance for the AV group when comparing between judgments of horizontal and vertical motion on the post-test (T(45) = -1.75, $p = 0.086$).

## Discussion

The present study shows it is possible to induce visual perceptual learning on a difficult, noisy, peripheral, global direction discrimination and integration task in virtual reality. In this context, we asked whether an audio-visual multisensory training approach could induce faster, greater, and/or spatially generalizable visual learning than training with a unisensory visual stimulus, as previously described by others who used a standard, 2D computer display, with external speakers generating the auditory stimuli [38,52,53].

After training concluded, the assessment of performance at trained locations revealed that both V and AV groups demonstrated learning that persisted into the post-test. In all conditions, there was a benefit from training that transferred spatially to the contralateral visual hemifield. The finding of transfer is similar to the finding of [54], although the latter only demonstrated transfer to a different quadrant in the same hemifield of vision. The generalizability indicated by spatial transfer of perceptual learning is generally considered an indication of a learned, higher-level representation that is location invariant [34]. For this reason, we interpret the successful transfer between visual hemifields observed presently as an indication that, for all participants, a component of perceptual learning occurred at higher levels of the visual processing hierarchy.

Although it was not surprising that the AV group demonstrated near-ceiling performance on horizontal judgments when training with an auditory stimulus that gave information about the horizontal direction, the effects this had on transfer were nuanced. When motion discrimination performance was assessed in the visual-only post-test at *trained* spatial locations, the AV group's ability to judge horizontal motion direction greatly exceeded their relatively modestly-improved ability to judge the vertical component of motion. These results lead us to speculate that the AV cue to horizontal motion direction may have additionally benefited from a distinct form of learning that was spatially specific. The retinotopic specificity of learning indicated by a failure of spatial transfer is generally interpreted as suggestive of low-level perceptual learning- i.e., at an earlier stage of visual processing [34,37,38]. Our results are consistent with the theory that low-level perceptual learning can be facilitated by multisensory stimuli, with the effect of raising the unisensory response above a threshold in neural activation [52,55]. Cross-modal interactions like these have been linked to the existence of direct connections between early stages of processing in the auditory and visual cortex

[56,57]. The alternative explanation that the presence of the auditory cue during training facilitated learning via endogenous, feature-based attention seems unlikely, both because of the timing of the cue in the present study (simultaneous with the visual stimulus), and because such attentional effects normally facilitate spatial transfer of learning [31,58], which was not observed here. Similarly, the alternative explanation that learning could be the result of a later-stage response-bias [59] does not offer a satisfactory account for the failure of transfer to the contralateral field.

In summary, both V and AV training groups demonstrated effects of perceptual learning as a result of training, which transferred to untrained portions of the visual field. While the AV group appeared to exhibit additional benefits of training with an auditory stimulus that provided unambiguous information about horizontal motion direction, these benefits were specific to the horizontal component of stimulus motion, and to the spatial location at which visual training occurred. We speculate that this occurred because the addition of the auditory component to the AV stimulus during training facilitated low-level perceptual learning, at an earlier stage of sensory processing.

Although the finding that participants were able to learn our difficult, direction discrimination and integration task in VR is consistent with results from a similar motion integration task performed on a conventional desktop computer display [37,38,60], participants in our tasks demonstrated lower learning rates. Choices related to experimental design prevented us from comparing the groups' learning rates when making judgments of horizontal motion direction. However, learning rates for the vertical motion component in our task were notably slower than those observed for horizontal motion in the studies by Levi, Seitz, and their colleagues. One possibility for the difference between our study and that of Seitz et al. is that they demonstrated a benefit for judgments of horizontal visual motion only when its movement direction was congruent with that of the auditory signal, whereas participants in our task made judgments of vertical motion that were not related to the motion direction of the auditory signal. There were other key task differences between the studies: notably, our task involved a 4-alternative-forced choice (AFC), whereas studies by Levi, Seitz and colleagues involved left/right 2AFC tasks. In addition, performing training in VR, while being mindful of holding both fixation and head position as steady as possible also rendered the task somewhat more cognitively and physically demanding. Task difficulty has been shown to have a strong effect on the quality of perceptual learning [16]. While most literature around task difficulty in perceptual learning centers around specificity of training, the results from [61] show shallower training slopes for a higher precision task than for the same task with lower precision. As task precision required for a 4AFC task is higher than for a left/right 2AFC task, a longer training period may have been needed to produce greater learning in our participants.

A related question that should be considered in this context is why the analysis of judgments of vertical motion direction thresholds during training indicate that our multisensory training methodology did not facilitate elevated learning rates for the AV group over the V group, as previously observed by [37] and [38]. The results in the former study offer particularly relevant insights as it also explored learning rates between V and AV groups making judgments of motion, as well as transfer of learning to a unisensory, visual-only post-test, albeit using a conventional display and computer speakers. Shams and Seitz [37] found that the auditory accompaniment to the visual motion stimulus facilitated faster learning rates only when the motion directions of the auditory and visual components were congruent. This was true even though the "incongruent" auditory stimulus' movement in the direction opposite to that of the visual stimulus was equally informative of the visual motion direction participants were asked to report. This suggests that the advantages of the congruent stimuli

are contingent on visual and auditory binding and causal inference of a common source, as described in [62]. This common-source inference was likely facilitated by the design choice to present visual and auditory components of the multisensory stimulus near each participant's sensory thresholds, facilitating judgments in a manner consistent with Bayesian integration [63]. In contrast, in the present study, AV participants' judgments of horizontal motion during training may have been driven almost entirely by the suprathreshold and unambiguous auditory stimulus, in a manner that likely dominated any Bayesian-style cue combination; this in turn, possibly led to causal inference of the vertical and auditory stimulus as arising from distinct sources. Consequently, in a manner similar to the incongruent condition of [37], analysis of judgments of vertical motion direction suggest that AV training in our experiment failed to elicit benefits on the rate of perceptual learning above those demonstrated by the V group.

Additionally, the rate of learning to judge the vertical component of motion was slightly lower for the AV group compared to those who trained with the unisensory visual stimulus. Although the magnitude of this effect was fairly weak ($\eta_p^2$ = 0.04), even the absence of an effect would differ from the findings of Seitz et al. [38], who also investigated the role of an auditory component on visual learning, and found auditory facilitation of visual perceptual learning [38]. We speculate that this can be explained by differences in task difficulty [16,61]: whereas the study by Seitz and colleagues used a task that remained at a fixed difficulty level, ours adaptively manipulated the level of difficulty through a staircase. Moreover, our task was quite difficult even at its lowest noise settings, causing almost a third of participants to fail the baseline titration tests described in section , causing them to be excluded from the rest of the study. In similar tasks completed during pilot testing, which only used a left/right 2AFC discrimination versus a 4AFC discrimination task, dropout levels were much lower, and almost exclusively due to self-exclusion due to discomfort in the VR headset. The high difficulty level in the present experiment was set deliberately – in part to encourage audiovisual interactions, which have been shown to be strengthened when visual information is of lower quality [64–66]. This raises the possibility that some of the benefits of auditory training might have been counteracted by the increased difficulty of the task.

We further speculate that the observed difference in learning rates between the V and AV groups when performing our relatively difficult task was also impacted by the allocation of attention during training, or through the reduction of uncertainty in the AV training condition. Consider that the auditory component was informative only about the horizontal component of motion and offered no advantage to judgments of vertical direction of motion. If the auditory cue resolved the horizontal component of motion, then the 4AFC visual motion direction judgment for the AV group was effectively reduced to two independent 2AFC tasks. While we did not observe the spatial transfer that characterizes some forms of attention-facilitated perceptual learning [31,58], it is likely that these tasks competed for attentional resources: one involving the perception of the suprathreshold auditory motion stimulus, and the other involving perception of the vertical component of visual motion in the stimulus. Thus, the presence of the auditory stimulus may have caused an increase in the allocation of attention towards resolving the motion direction of the auditory stimulus at the cost of resolving the vertical component of motion. Although attention can be a strong driver of visual perceptual learning [67,68], it is also a limited cognitive resource, and the demands that audiovisual integration places on attention are especially strong when the auditory and visual stimuli are spatially and temporally coherent [69–71]. Moreover, information about the horizontal component of motion was irrelevant to correctly judging the vertical component, which was the only dimension along which relative learning rates could be directly compared across groups. These observations suggest that the cross-modal spread of attention may have led

to the auditory sub-task capturing some of the participant's attention away from the visual judgment of the vertical direction of motion [72,73].

The present study is a demonstration of the feasibility of implementing multisensory perceptual learning in virtual reality, and builds upon similar demonstration by Alwashmi et al. [41] with regards to a visual/auditory scanning task. It is notable that in the present study, six out of twenty-three participants attended the initial practice block, but opted out of continuing the study for the additional eleven consecutive days, leaving a final participant pool of 17 individuals. None of the participants reported simulator sickness, which is unsurprising given the sparsity of the visual environment and the lack of full-body translation. Instead, we attribute this high dropout rate to the difficulty of maintaining fixation, and head position, as well as to the tedious nature of the motion discrimination task. These limitations may be addressed in future studies simply by adopting tasks that leverage what virtual-reality displays have to offer: the ability to design engaging, visually-guided tasks that are parameterized to facilitate the investigation of perceptual thresholds under systematically changing visual conditions.

In conclusion, we hereby describe the impact of adding a congruent, auditory component to a difficult, visual perceptual learning task performed in the visual periphery. We found that participants deployed attention across both sensory modalities, and used all the information provided to perform a noisy, global motion discrimination task. Our results suggest that sound can facilitate visual perceptual learning, but that the results may not transfer across dimensions (i.e., direction of motion) or locations in the visual field. Multisensory engagement is one factor among many, which can impact learning. We also posit that both task-related and unrelated factors – such as the physical and cognitive demands of using a particular technology like VR – must be carefully tuned if the goal is to optimize the rate of sensory learning.

## Author contributions

**Conceptualization:** Melissa J. Polonenko, Krystel R. Huxlin, Gabriel J. Diaz.

**Data curation:** Catherine A. Fromm, Gabriel J. Diaz.

**Formal analysis:** Catherine A. Fromm, Gabriel J. Diaz.

**Funding acquisition:** Gabriel J. Diaz.

**Investigation:** Catherine A. Fromm, Ross K. Maddox, Krystel R. Huxlin, Gabriel J. Diaz.

**Methodology:** Catherine A. Fromm, Ross K. Maddox, Melissa J. Polonenko, Krystel R. Huxlin, Gabriel J. Diaz.

**Project administration:** Gabriel J. Diaz.

**Resources:** Gabriel J. Diaz.

**Software:** Catherine A. Fromm, Gabriel J. Diaz.

**Supervision:** Ross K. Maddox, Melissa J. Polonenko, Krystel R. Huxlin, Gabriel J. Diaz.

**Validation:** Catherine A. Fromm, Gabriel J. Diaz.

**Visualization:** Catherine A. Fromm, Melissa J. Polonenko, Krystel R. Huxlin, Gabriel J. Diaz.

**Writing – original draft:** Catherine A. Fromm, Gabriel J. Diaz.

**Writing – review & editing:** Catherine A. Fromm, Ross K. Maddox, Melissa J. Polonenko, Krystel R. Huxlin, Gabriel J. Diaz.

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
