## [Decision Letter · Decision Letter 0]

2 Sep 2024

PONE-D-24-31083Multisensory stimuli facilitate low-level perceptual learning on a difficult global motion task in virtual realityPLOS ONE

Dear Dr. Huxlin,

Thank you for submitting your manuscript to PLOS ONE. After careful consideration, we feel that it has merit but does not fully meet PLOS ONE’s publication criteria as it currently stands. Therefore, we invite you to submit a revised version of the manuscript that addresses the points raised during the review process.

Your manuscript has been reviewed by two experts in the topic. As you will see below, while both reviewers were generally positive about your work, they also raised a number of concerns about the experimental design and the interpretation of the data which should be addressed in your revision.

We look forward to receiving your revised manuscript.

Kind regards,

Patrick Bruns

Academic Editor

PLOS ONE

Journal Requirements:

1. When submitting your revision, we need you to address these additional requirements. Please ensure that your manuscript meets PLOS ONE's style requirements, including those for file naming. The PLOS ONE style templates can be found at https://journals.plos.org/plosone/s/file?id=wjVg/PLOSOne_formatting_sample_main_body.pdf and https://journals.plos.org/plosone/s/file?id=ba62/PLOSOne_formatting_sample_title_authors_affiliations.pdf 2. We note that the grant information you provided in the ‘Funding Information’ and ‘Financial Disclosure’ sections do not match.  When you resubmit, please ensure that you provide the correct grant numbers for the awards you received for your study in the ‘Funding Information’ section. 3. Thank you for stating the following financial disclosure: "Research reported in this publication was supported by a grant from the National Eye Institute of the National Institutes of Health under award number 1R15EY031090". Please state what role the funders took in the study.  If the funders had no role, please state: "The funders had no role in study design, data collection and analysis, decision to publish, or preparation of the manuscript." If this statement is not correct you must amend it as needed. Please include this amended Role of Funder statement in your cover letter; we will change the online submission form on your behalf. 4. Thank you for stating the following in the Acknowledgments Section of your manuscript: "Research reported in this publication was supported by the National Eye Institute of the National Institutes of Health under award number 1R15EY031090." We note that you have provided funding information that is not currently declared in your Funding Statement. However, funding information should not appear in the Acknowledgments section or other areas of your manuscript. We will only publish funding information present in the Funding Statement section of the online submission form. Please remove any funding-related text from the manuscript and let us know how you would like to update your Funding Statement. Currently, your Funding Statement reads as follows: "Research reported in this publication was supported by a grant from the National Eye Institute of the National Institutes of Health under award number 1R15EY031090". Please include your amended statements within your cover letter; we will change the online submission form on your behalf.

Reviewers' comments:

Reviewer's Responses to Questions

**Comments to the Author**

1. Is the manuscript technically sound, and do the data support the conclusions?

Reviewer #1: Yes

Reviewer #2: Partly

2. Has the statistical analysis been performed appropriately and rigorously? 

Reviewer #1: Yes

Reviewer #2: Yes

3. Have the authors made all data underlying the findings in their manuscript fully available?

Reviewer #1: Yes

Reviewer #2: Yes

4. Is the manuscript presented in an intelligible fashion and written in standard English?

Reviewer #1: Yes

Reviewer #2: Yes

5. Review Comments to the Author

Reviewer #1: The authors investigated the differences between a visual-only and an audio-visual motion direction discrimination perceptual learning task with a head-mounted display (HMD). The visual-only group consisted of 8, and the audio-visual group of 9 healthy young adults. Training on a global motion task was performed on 10 days. In the audio-visual group the visual motion was accompanied by a congruent virtual horizontally moving sound. In addition, visual-only tests were conducted for both groups before and after the training. The authors found that performance (direction range threshold) increased across the 10 training sessions in both groups. Mean performance (but not learning rate) during training was enhanced in the audio-visual group for horizontal but not for vertical motion discrimination. Similarly, at the visual-only post-test the audio-visual group showed enhanced horizontal discrimination performance (similar to the last training session), but not vertical discrimination performance. The authors conclude that auditory signals may enhance visual perceptual learning at releatively low levels of processing.

The study is relevant as it addresses the question whether and how visual perceptual learning can be enhanced by multisensory stimulation. The approach is solid. The results are plausible and largely consistent with prevous findings. The chosen task and display are novel, interesing, and relevant. As of this, the study would elaborate and strengthen the existing literature on this topic. However, it is not the first study to address this issue. Some previous studies were cited, but some still need to be discussed. The sample size is small, but acceptable for this type of research. Some analyses are somewhat inconclusive and could be strengthened and simplified.

a) Major concerns

l. 34ff: Some relevant literature is missing. Previous work found that sounds may facilitate visual perceptual learning at visual apertures that spatially overlap with the sound source (dx.doi.org/10.1007/s00221-009-1769-6). The audio-visual alignment may be partially altered by perceptual learning (10.3758/s13415-010-0006-x). Auditory cues may augment brain activation changes and enhance performance gains in a virtual reality visual scanning training (dx.doi.org/10.1016/j.neuroimage.2023.120483).

l. 132: What was the convergence level of the 2-up-1-down staircase? Please add. Maybe this literature helps (Levitt H. Transformed up-down methods in psychoacoustics. J Acoust Soc Am. 1971 Feb;49(2):Suppl 2:467+.).

l. 197: What was the criterion for fixation? How much deviation was allowed? Please add.

l. 248: The phrase 'which was statistically independent' is not correct. As both judgements (vertical and horizontal) were done by the same person at the same trial, it is very unlikely and cannot be assumed without testing that the judgements were fully statistically independent. Although I understand that the judgements were along a different (orthogonal) dimension and that it makes sense to analyze these dimensions separately, the authors should phrase this more carefully. Moreover, it might be risky to include motion component (global, vertical, horizontal) in the same statistical model (maybe not done, but inconclusive at l. 267) because of possible statistical dependencies (without controlling for it). Conducting separate analyses for each of the motion components is preferrable.

l. 264ff: Why were the effects tested with a linear mixed-effects model? The lme4 package is usually used for multi-level analysis. However, this study is a simple-level design with only fixed-effects (except for the subjects variability). Although it is not wrong to apply lme4, simpler and more intuitive linear models (e.g., 'lm', 'sem') would also work here. Similarly, why were the comparisons regarding the post-test performance (l. 310ff) not tested by simpler methods such as ANOVAs and t-tests? Again, lme4 is not wrong, but less intuitive. The R package 'rstatix' would provide easy-to-use functions (e.g., res.aov, get_anova_table).

l. 274: The statement 'learned at a slightly faster rate than' is not justified by the results and is misleading, because no signifiant interaction was observed for horizontal judgements. It would be more adequate to state 'learned at a similar rate than'. If the statement refers to vertical judgements, this should be made clearer.

l. 281ff: The reported statistics are very sparse. Only p-values are provided. No t-values, degrees of freedom (or maybe even effect sizes) are provided. These statistical values should also be reported.

l. 281ff (or elsewhere): Did the authors check and compare response times? Response times might help to resolve the ceiling issue on horizontal judgments in the audio-visual group? They might also indicate (or rule out) a speed-accuracy trade-off.

l. 281ff (or elsewhere): Did the authors check and compare eye movement patterns? Although participants were instructed to fixate and that was assured by monitoring eye movements, there may still be small eye movements. At least there should be a statement about the extent of eye movements. If there are group differences, these should be reported as they could provide meaningful information. For instance: If the authors are right that in the visual-only group (compared to the audio-visual group) learning occured at later stages of processing, it might be possible to see this in small but significant differences in eye movement patterns.

l. 310ff: Why did the authors only compare post-test performance with the last-day of learning? Why not a pre-test to post-test comparison? If I am not mistaken, the pre-test was similar to the post-test. As the pre-test would be the same in both groups (unlike the last training session), it would provide a less confounded comparison between the visual-only and the audio-visual group. Even if pre-tests cannot be directly compared with post-test (in case they were different tasks), more results (e.g., direction range thresholds, response times) of the pre-tests should be reported in order to rule out pre-training group differences.

l. 399 - l. 401: The authors should elaborate their argument and integrate (consistent) previous findings. The argument that low-level multisensory interactions promote perceptual learning is consistent with previous findings (dx.doi.org/10.1007/s00221-009-1769-6). It also is supported by brain imaging studies showing functional (dx.doi.org/10.1002/hbm.20560) and structural connections (dx.doi.org/10.1007/s00221-011-2715-y) between the auditory and visual cortex.

l. 448ff: I am not sure whether the lower vertical motion learning rate in the AV group differs from that of Seitz et al. (2006, as cited). Seitz et al. examined horizontal motion and not vertical motion. It seems like the main difference is between horizontal motion (congruent to auditory signal) and vertical motion (not related to auditory signal).

b) Minor concerns

l. 40: What does it mean 'when one of the sensory modalities is compromised'? Please specify, add reference, or drop.

l. 42: 'intact individuals' sounds quite technical. Maybe a more respectful term can be used such as 'healthy individuals'.

l. 66ff: The size for each group (visual-only, audio-visual) should be mentioned here as it does not appear anywhere else in the main text. Did the authors record nicotine consumption of the participants as nicotine may strongly modulate perceptual learning via the cholinergic system (dx.doi.org/10.1016/j.neuropharm.2012.06.019)? If so, please report.

Fig. 3: What is the meaning of the error bars? Standard error of the mean (SEM)? Please clarify in the caption. Maybe the legend inset can be modified in a way that it becomes clear what symbol (disc vs. triangle) and what line (solid vs. dashed) refers to the V or AV group. Using only colors is not optimal when printed in grayscale. Right now it is necessary to consult the caption.

Fig. 4: What is the meaning of the boxes, the lines, and the dots? Box-Whisker-plots? Please clarify in the caption. If dots are outliers that were excluded from the analyses, it might be a serious problem that for some crucial comparsions 2 out of 8 data points were excluded. If so, this needs to be addressed. Given that the variables were quantitative, why was no standard bar graph with error bars used? Pair-wise t-tests could be performed and the results added to the graph. The chosen colors are hard to distinguish when printed in grayscale.

Reviewer #2: In this study, authors examined motion direction discrimination training in a virtual reality (VR) environment. Participants were divided into two groups and trained for 10 days. The key experimental manipulation was the presence of an auditory cue coherent with the horizontal motion direction of the stimuli, provided to one group (Audio video [AV] group). The results showed learning effects in both groups, with the group that did not receive the auditory cue (Video [V] group) exhibiting a slightly faster learning rate. Post-tests to assess the generalization of learning indicated transfer to an untrained retinal quadrant, primarily in the group without the auditory cue.

I find this study and its setup both interesting and timely, with significant potential for advancing our understanding of visual processing and learning in innovative ways. However, I have concerns about the experimental design, the interpretation of the data and the structure of the discussion section.

Concerning the experimental design, I'm unsure why the authors used interleaved staircases and separated the horizontal and vertical components in a task where participants were asked to report the diagonal direction of moving stimuli. My understanding is that participants provided responses like 'north-west' or 'south-east,' and, depending on which staircase was active, only the horizontal or the vertical component was then fed to the staircase. However, the actual stimulus direction was 45° away from both horizontal and vertical.

I might be missing something, but I don’t understand why the authors didn’t design the paradigm with a direct 1:1 correspondence between the auditory cue and the motion direction. For example, instead of using diagonal motions, they could have used cardinal directions (horizontal versus vertical). This is similar to what Seitz et al. (2009), whom the authors cite as a reference for multisensory facilitation in perceptual learning, did. Crucially, Seitz et al. (2009) state ‘Notably, semantic congruency was required for enhanced visual recognition. In our experiment, audio-visual directions were congruent; it remains to be explored whether congruency is required, or whether any type of sound can aid learning’.

Concerning the experimental design, I believe that the mechanisms invoked by the authors might not be the ones (or the only ones) at play here. With the paradigm used, it is possible that the acoustic cue in the AV did not act as a multisensory facilitation stimulus, but rather as a cue to reduce uncertainty. Once the participants in the AV group identified the direction of the sound cue, they could narrow down the possible responses from 4 to 2. Thus, the two groups might have performed two qualitatively different tasks: the V group engaged in a 4AFC, while the AV group might have performed a two-step task, with processing of the cue, and then determining whether the movement was diagonally up or down.

Multisensory research has long examined how the choice of paradigm can affect whether an improvement in performance due to multisensory presentation truly reflects s early-stage sensory interaction or is a result of later-stage response bias effect (i.e., Odgaard et al. (2003). The latter hypothesis is consistent with the statement on line 325: ‘all groups demonstrated persistence of learning from t10 to the post-test, albeit at a slightly reduced level for AV group’s judgments of horizontal motion direction, which is not surprising given that the suprathreshold auditory cue present for t10 was no longer available for the post-test’ and examination of Figure 4. This statement seems to suggest that the improvement observed in the training was not due to multisensory facilitation, but possibly reflected the use of the acoustic cue as a reduction of spatial uncertainty in judging motion direction.

Concerning the interpretation of the data, when addressing the lack of retinal transfer in the post tests of the AV group, the authors speculate that ‘this occurred because the addition of the auditory component to the AV stimulus during training facilitated low-level perceptual learning at an earlier stage of sensory processing’, while, for the V group, ‘learning occurred at higher levels in the visual processing hierarchy’, and it was ‘location invariant’.

I am not particularly convinced by the suggestion that, quite counterintuitively, multisensory training would have led to larger retinal specificity than unisensory training. This seems to be at odds with most literature on multisensory learning (i.e., Shams and Seitz, 2008; Paraskevopoulos et al., 2024) and models of perceptual learning. Similarly, I’m not really convinced by the use of ‘difficult’ to describe the task here. On one hand, difficulty was kept constant by using an adaptive procedure that targeted a specific threshold, thus constantly adapting to the participant’s performance. In the experimental procedures the authors further explain how they conducted pre-tests to calibrate the difficulty of the task to each participant’s ability (i.e., line 172-178: ‘If the practice block was so difficult that the participants could not progress, or so easy that every trial was correct, the proportion of randomly-moving dots was adjusted in increments of 10% noise dots until the participant was able to maintain a direction range threshold of at least 80◦ within the first 20 trials. Following this coarse adjustment of the baseline signal:noise dot ratio, participants performed a titration block of 100 trials. Performance in this block was used to fine-tune the signal:noise dot ratio for the rest of the experiment.’). On the other hand, some of the theoretical framework the authors have referred to in the context of transfer of learning, and level at which perceptual learning takes place, would suggest that, if anything, a difficult task would lead to larger specificity and less retinal transfer of learning. Thus, it’ unclear if the authors set out the task to be purposely ‘difficult’, expecting the multisensory modulation to counteract the expected specificity that derives from a ‘difficult’ task (i.e., Huang and Seitz, 2014), or whether this consideration was a posteriori.

Additionally, in terms of neural mechanisms, the evidence that, unlike other multisensory stimulation studies, here the multisensory learning did not transfer to other location might suggest that judging diagonal directions of motion may not entail extracting horizontal and vertical cardinal directions, but rather judging movements along the diagonal.

In other words, judging diagonal motion direction may not be a product of independently judging the horizontal and vertical, 90° apart, components of said motion. The participants might have learned to associate the horizontal direction of the sound with the horizontal component of motion (hence the higher post-training performance in the transfer tasks), but this did not provide an advantage in performing the training task, which implied discriminating the diagonal direction. A more congruent mapping of the sound with the motion direction (i.e., horizontal and vertical sounds and horizontal or vertical, rather than diagonal, motion direction) might have provided the multisensory advantage the authors were looking for, similar to what Seitz et al. (2009) reported.

Concerning the discussion section, I believe it might benefit from a restructuring and rewording to improve clarity.

This section begins with the statement on line 366 ‘Moreover, for all conditions except the AV group’s judgments of horizontal motion direction, the elevated thresholds developed during training transferred spatially to the contralateral visual hemifield’. Few lines above this, the authors write ‘Visual inspection of Fig 4B suggests that the AV group’s elevated performance levels did not transfer to untrained locations’, with a sub header stating: ‘The benefit from AV training was spatially specific’.

The discussion continues commenting on the level at which learning might have taken place, suggesting that ‘The generalizability indicated by spatial transfer of perceptual learning is generally considered an indication of a learned higher-level representation that is location invariant’.

The next sentence reads: “For this reason, we interpret […] for all participants, perceptual learning occurred at higher levels in the visual processing hierarchy.

However, this specification ‘for all participants’ is not consistent with the results in figure 4, with the aforementioned sub header, and with the authors’ own comments on the results (‘AV group’s elevated performance levels did not transfer to untrained location’)

This is, somehow anti-climatically, addressed in the following sentence: ‘There was, however, one notable condition in which spatial transfer did not occur: that of the AV group’.

This is rather confusing, as more than ‘one notable condition’, this represents the main experimental manipulation (out of only two), and the one prominently mentioned in the title of the paper.

Overall, I commend the authors for pushing the envelope of psychophysical experiments and I sympathize with the difficulty that comes with the interpretation of less-than-straightforward results, but I invite caution in doing so. I appreciated the second part of the discussion and I found that some of my previous comments have been already put forward, however it is important to recognize possible limitations of the experimental setup and of the results. For example, the emphasis on ‘multisensory’ (and in part ‘difficult’) in the title might not be reflective of the main result of the paper.

Minor comments

While I believe VR holds the promise of liberating vision science from the constraints of a lab setting, it might be worth briefly discussing some of the downside of this setup, mostly in terms of possible discomfort and compliance issues, particularly in relation to line 75-78: ‘Six participants (3 female) attended the first session (practice block) only. They either opted out of the study due to discomfort using the virtual reality system or they were excluded by the experimenter because of unusually poor task performance in the practice block. ‘

Line 20-24: maybe worth mentioning alternative hypotheses concerning the mechanisms of VPL, such as the sharpening of the tuning curves of neurons in early sensory areas (i.e., Ghose et al., Schoups et al.).

You might want to move the explanation on Line 189 (Participants were divided into two groups […] hereafter referred to as the V and AV groups) a bit earlier, as you already use the acronyms V and AV before you explicitly define them.

Line 197: ‘After the eye tracker recorded 1s of fixation on the target’. maybe you can replace ‘target’ with ‘fixation cross’ to avoid confusion with the actual stimulus.

line 91-92: ‘At the start of each testing or training session, participants adjusted the headset to

improve comfort and display quality’. Would this adjustment have affected the viewing distance or eccentricity of the stimuli?

It would have been interesting, both from a practical and theoretical point of view, to see a comparison between performance in VR and performance on a ‘classic’ psychophysical setup, or even just post tests conducted on a standard lab setup to characterize generalization across setups.

6. PLOS authors have the option to publish the peer review history of their article (what does this mean?). If published, this will include your full peer review and any attached files.

Reviewer #1: No

Reviewer #2: No

---

## [Author Response · Author response to Decision Letter 1]

19 Nov 2024

We thank the reviewers and editors for their careful evaluation of our manuscript and their many insightful suggestions and questions, which we address in a point-by-point manner below. Changes in the manuscript have been denoted by blue text. We believe that the manuscript has been significantly improved with these changes and hope that it meets with your approval.

Response: Done

Response: Our Financial Disclosure section has been revised as follows, to be consistent with the grants listed in the Funding Information section:

Financial disclosure: Research reported in this publication was supported by a grant from the National Eye Institute of the National Institutes of Health under award number R15 EY031090. KRH’s work was also partially supported by an Unrestricted Grant from Research to Prevent Blindness to the Flaum Eye Institute at the University of Rochester. The funders had no role in study design, data collection and analysis, decision to publish, or preparation of the manuscript.

3. Thank you for stating the following financial disclosure: "Research reported in this publication was supported by a grant from the National Eye Institute of the National Institutes of Health under award number 1R15EY031090". Please state what role the funders took in the study. If the funders had no role, please state: "The funders had no role in study design, data collection and analysis, decision to publish, or preparation of the manuscript."

Response: Please see response above - the new Financial Disclosure section now includes the required statement.

Response: We have also amended the Cover Letter as requested with the new Financial Disclosure statement.

4. Thank you for stating the following in the Acknowledgments Section of your manuscript: "Research reported in this publication was supported by the National Eye Institute of the National Institutes of Health under award number 1R15EY031090."

We note that you have provided funding information that is not currently declared in your Funding Statement. However, funding information should not appear in the Acknowledgments section or other areas of your manuscript. We will only publish funding information present in the Funding Statement section of the online submission form. Please remove any funding-related text from the manuscript and let us know how you would like to update your Funding Statement. Currently, your Funding Statement reads as follows: "Research reported in this publication was supported by a grant from the National Eye Institute of the National Institutes of Health under award number 1R15EY031090".

Response: We have deleted the Acknowledgements.

Response: Done - thank you! 

Reviewer #1: The authors investigated the differences between a visual-only and an audio-visual motion direction discrimination perceptual learning task with a head-mounted display (HMD). The visual-only group consisted of 8, and the audio-visual group of 9 healthy young adults. Training on a global motion task was performed on 10 days. In the audio-visual group the visual motion was accompanied by a congruent virtual horizontally moving sound. In addition, visual-only tests were conducted for both groups before and after the training. The authors found that performance (direction range threshold) increased across the 10 training sessions in both groups. Mean performance (but not learning rate) during training was enhanced in the audio-visual group for horizontal but not for vertical motion discrimination. Similarly, at the visual-only post-test the audio-visual group showed enhanced horizontal discrimination performance (similar to the last training session), but not vertical discrimination performance. The authors conclude that auditory signals may enhance visual perceptual learning at relatively low levels of processing.

The study is relevant as it addresses the question whether and how visual perceptual learning can be enhanced by multisensory stimulation. The approach is solid. The results are plausible and largely consistent with previous findings. The chosen task and display are novel, interesting, and relevant. As of this, the study would elaborate and strengthen the existing literature on this topic. However, it is not the first study to address this issue. Some previous studies were cited, but some still need to be discussed. The sample size is small, but acceptable for this type of research. Some analyses are somewhat inconclusive and could be strengthened and simplified.

a) Major concerns

l. 34ff: Some relevant literature is missing. Previous work found that sounds may facilitate visual perceptual learning at visual apertures that spatially overlap with the sound source (dx.doi.org/10.1007/s00221-009-1769-6). The audio-visual alignment may be partially altered by perceptual learning (10.3758/s13415-010-0006-x). Auditory cues may augment brain activation changes and enhance performance gains in a virtual reality visual scanning training (dx.doi.org/10.1016/j.neuroimage.2023.120483).

Response: We thank the reviewer for the supporting references which have been incorporated into the Introduction and Discussion sections of the manuscript.

l. 132: What was the convergence level of the 2-up-1-down staircase? Please add. Maybe this literature helps (Levitt H. Transformed up-down methods in psychoacoustics. J Acoust Soc Am. 1971 Feb;49(2):Suppl 2:467+.).

Response: The convergence level of the staircase (70.7%) is now reported in the section Methods:Apparatus:Motion Discrimination Task.

l. 197: What was the criterion for fixation? How much deviation was allowed? Please add.

Response: This information was provided towards the end of the third paragraph of the Methods:Apparatus section (around line 122): “Trials would not begin unless fixation was within 0.3 degrees of the fixation point, and head pose was within 2 degrees of the world-fixed box for one uninterrupted second. The trial was aborted if gaze deviated by more than 1.5 degrees from fixation mid-trial, if head orientation deviated by more than 2 degrees from the designated pose, or if position deviated by more than 10 inches from the position defined by the resting pose.”

l. 248: The phrase 'which was statistically independent' is not correct. As both judgements (vertical and horizontal) were done by the same person at the same trial, it is very unlikely and cannot be assumed without testing that the judgements were fully statistically independent. Although I understand that the judgements were along a different (orthogonal) dimension and that it makes sense to analyze these dimensions separately, the authors should phrase this more carefully. Moreover, it might be risky to include motion component (global, vertical, horizontal) in the same statistical model (maybe not done, but inconclusive at l. 267) because of possible statistical dependencies (without controlling for it). Conducting separate analyses for each of the motion components is preferrable.

Response: Thank you. We have now clarified the statement in the third paragraph of the Methods:Analysis section (around line 258).

l. 264: Why were the effects tested with a linear mixed-effects model? The lme4 package is usually used for multi-level analysis. However, this study is a simple-level design with only fixed-effects (except for the subjects variability). Although it is not wrong to apply lme4, simpler and more intuitive linear models (e.g., 'lm', 'sem') would also work here. Similarly, why were the comparisons regarding the post-test performance (l. 310ff) not tested by simpler methods such as ANOVAs and t-tests? Again, lme4 is not wrong, but less intuitive. The R package 'rstatix' would provide easy-to-use functions (e.g., res.aov, get_anova_table).

Response: Whereas use of a mixed ANOVA would allow us to test for categorical differences, it does not provide an estimate of the slope. In contrast, linear mixed effects models (LMM) are an extension of linear regression that provide coefficients for both fixed and random effects. This in turn, provides insight into relative learning rates during training compared to a simple ANOVA. We chose to continue with LMM for other measures for the sake of internal consistency within the manuscript, and because, as the reviewer pointed out, LMM are uniquely able to take into account the subject variability to reveal the main fixed effects in the case of repeated measures designs.

l. 274: The statement 'learned at a slightly faster rate than' is not justified by the results and is misleading, because no significant interaction was observed for horizontal judgements. It would be more adequate to state 'learned at a similar rate than'. If the statement refers to vertical judgements, this should be made clearer.

Response: The reviewer is correct - we apologize for this error. We have now updated the text accordingly.

l. 281ff: The reported statistics are very sparse. Only p-values are provided. No t-values, degrees of freedom (or maybe even effect sizes) are provided. These statistical values should also be reported.

Response: We have revised the Results section to include the requested information, and described the associated methods in the section Methods:Statistics.

l. 281ff (or elsewhere): Did the authors check and compare response times? Response times might help to resolve the ceiling issue on horizontal judgments in the audio-visual group? They might also indicate (or rule out) a speed-accuracy trade-off.

Response: Unfortunately, response times were not recorded, nor were subjects instructed to make speeded judgments

l. 281ff (or elsewhere): Did the authors check and compare eye movement patterns? Although participants were instructed to fixate and that was assured by monitoring eye movements, there may still be small eye movements. At least there should be a statement about the extent of eye movements. If there are group differences, these should be reported as they could provide meaningful information. For instance: If the authors are right that in the visual-only group (compared to the audio-visual group) learning occurred at later stages of processing, it might be possible to see this in small but significant differences in eye movement patterns.

Response: Given the levels of accuracy and precision of integrated eye trackers in VR, we hesitate to draw conclusions from eye movements smaller than the 0.6 degree window of restriction used to enforce fixations. That being said, the range of eye movement deviation was not recorded. Quantitative information on the accuracy and precision of the Vive Pro is available here: https://www.ncbi.nlm.nih.gov/pmc/articles/PMC10136368/

l. 310ff: Why did the authors only compare post-test performance with the last-day of learning? Why not a pre-test to post-test comparison? If I am not mistaken, the pre-test was similar to the post-test. As the pre-test would be the same in both groups (unlike the last training session), it would provide a less confounded comparison between the visual-only and the audio-visual group. Even if pre-tests cannot be directly compared with post-test (in case they were different tasks), more results (e.g., direction range thresholds, response times) of the pre-tests should be reported in order to rule out pre-training group differences.

Response: Thank you for the suggestion. Unfortunately, participants needed considerable training and coaching to correctly perform the task at hand during their first day in the lab. As such, much of their first day in the setup was spent adjusting the helmet, teaching them to perform the task while keeping their head steady and testing a range of stimulus parameters in preparation for selecting those to be used for the actual training phase, starting the next day. As such, no proper pre-test measures were actually collected as performance was extremely variable between participants during this training/titration phase, and also involved different numbers of trials, depending on how easily a participant adapted to the VR system and the task. As such, we do not have an easy way to compare baseline data during this initial visit between participants.

l. 399 - l. 401: The authors should elaborate their argument and integrate (consistent) previous findings. The argument that low-level multisensory interactions promote perceptual learning is consistent with previous findings (dx.doi.org/10.1007/s00221-009-1769-6). It also is supported by brain imaging studies showing functional (dx.doi.org/10.1002/hbm.20560) and structural connections (dx.doi.org/10.1007/s00221-011-2715-y) between the auditory and visual cortex.

Response: We thank the reviewer for the suggested references. We have added them and the requested additional discussion supporting our conclusions around line 417.

l. 448ff: I am not sure whether the lower vertical motion learning rate in the AV group differs from that of Seitz et al. (2006, as cited). Seitz et al. examined horizontal motion and not vertical motion. It seems like the main difference is between horizontal motion (congruent to auditory signal) and vertical motion (not related to auditory signal).

Response: We agree with the reviewer and have modified the related text in the discussion section accordingly.

b) Minor concerns

l. 40: What does it mean 'when one of the sensory modalities is compromised'? Please specify, add reference, or drop.

Response: We have dropped this statement from the manuscript.

l. 42: 'intact individuals' sounds quite technical. Maybe a more respectful term can be used such as 'healthy individuals'.

Response: Agreed and changed.

l. 66ff: The size for each group (visual-only, audio-visual) should be mentioned here as it does not appear anywhere else in the main text.

Response: We have added the relevant information to the Participants section.

Did the authors record nicotine consumption of the participants as nicotine may strongly modulate perceptual learning via the cholinergic system (dx.doi.org/10.1016/j.neuropharm.2012.06.019)? If so, please report.

Response: We did not record nicotine consumption.

Fig. 3: What is the meaning of the error bars? Standard error of the mean (SEM)? Please clarify in the caption.

Response: We have clarified the captions accordingly.

Maybe the legend inset can be modified in a way that it becomes clear what symbol (disc vs. triangle) and what line (solid vs. dashed) refers to the V or AV group. Using only colors is not optimal when printed in grayscale. Right now it is necessary to consult the caption.

Response: We have adjusted the legend accordingly.

Fig. 4: What is the meaning of the boxes, the lines, and the dots? Box-Whisker-plots? Please clarify in the caption. If dots are outliers that were excluded from the analyses, it might be a serious problem that for some crucial comparisons 2 out of 8 data points were excluded. If so, this needs to be addressed. Given that the variables were quantitative, why was no standard bar graph with error bars used?

Response: No data points were excluded from analysis. The requested information has been added to the caption of Figure 4.

Pairwise t-tests could be performed and the results added to the graph.

Response: W

---

## [Decision Letter · Decision Letter 1]

16 Dec 2024

PONE-D-24-31083R1Multisensory stimuli facilitate low-level perceptual learning on a difficult global motion task in virtual realityPLOS ONE

Dear Dr. Huxlin,

Thank you for submitting your manuscript to PLOS ONE. After careful consideration, we feel that it has merit but does not fully meet PLOS ONE’s publication criteria as it currently stands. Therefore, we invite you to submit a revised version of the manuscript that addresses the points raised during the review process.

As you will see below, both reviewers were happy with your revisions. However, Reviewer #1 pointed our a few remaining minor issues, including the color coding in Fig. 4, which should be corrected. 

We look forward to receiving your revised manuscript.

Kind regards,

Patrick Bruns

Academic Editor

PLOS ONE

Journal Requirements:

Reviewers' comments:

Reviewer's Responses to Questions

**Comments to the Author**

1. If the authors have adequately addressed your comments raised in a previous round of review and you feel that this manuscript is now acceptable for publication, you may indicate that here to bypass the “Comments to the Author” section, enter your conflict of interest statement in the “Confidential to Editor” section, and submit your "Accept" recommendation.

Reviewer #1: (No Response)

Reviewer #2: All comments have been addressed

2. Is the manuscript technically sound, and do the data support the conclusions?

Reviewer #1: Yes

Reviewer #2: Yes

3. Has the statistical analysis been performed appropriately and rigorously? 

Reviewer #1: Yes

Reviewer #2: Yes

4. Have the authors made all data underlying the findings in their manuscript fully available?

Reviewer #1: Yes

Reviewer #2: Yes

5. Is the manuscript presented in an intelligible fashion and written in standard English?

Reviewer #1: Yes

Reviewer #2: Yes

6. Review Comments to the Author

Reviewer #1: The authors have adressed most of my concerns or explained why not. I still have some minor issues though (primarily editorial):

Abstract: "thanunisensory" -> "than unisensory"

Abstract: "specificity suggested" -> "specificity suggests"

l. 43: "intact individuals" -> "healthy individuals"

l. 259: "was statistically independent" -> "was independent"

l. 308: "))" -> ")"

p. 8, line 302 to p. 10, line 359: Use equal sign for exact p values (e.g., "p = .290") and the smaller sign only for p values that cannot be reported exactly because of too many decimals (e.g., "p < .001").

Caption of Fig. 4 (p. 9) and 2nd para of p. 10: The bars in Fig. 4 are actually red and blue and not orange and purple. Please make consistent either by correcting the text or the colors in Figure 4.

Reviewer #2: I commend the authors for their effort in successfully addressing my comments. I am happy with the current version of the manuscript.

7. PLOS authors have the option to publish the peer review history of their article (what does this mean?). If published, this will include your full peer review and any attached files.

Reviewer #1: No

Reviewer #2: **Yes: **Marcello Maniglia

---

## [Author Response · Author response to Decision Letter 2]

20 Jan 2025

We thank you for your significant efforts over the course of the review process, and the positive impact they have had on the final manuscript. The final changes suggested were minor, and are reflected in the revised manuscript with blue text, as follows:

Reviewer #1:

Abstract: "specificity suggested" -> "specificity suggests"

Response: done

l. 43: "intact individuals" -> "healthy individuals"

Response: done

l. 259: "was statistically independent" -> "was independent"

Response: done

l. 308: "))" -> ")"

Response: done

p. 8, line 302 to p. 10, line 359: Use equal sign for exact p values (e.g., "p = .290") and the smaller sign only for p values that cannot be reported exactly because of too many decimals (e.g., "p < .001").

Response: done (but not colored blue)

However, there are a few changes we did not make:

● Reviewer #1 reported a typographical error in the abstract (“thanunisensory”) that was not present in our version, or the version provided by the PLOS automated formatting system. As such, we are unable to address this issue.

● Reviewer #1: Caption of Fig. 4 (p. 9) and 2nd para of p. 10: The bars in Fig. 4 are actually red and blue and not orange and purple. Please make it consistent either by correcting the text or the colors in Figure 4.

This was not a mistake - we surmise it may have been a perceptual differences caused by differences in display properties, a color-space conversion problem, the heavy image compression used in generation of the PLOS preprint, or other factors. After some deliberation with colleagues, we have changed the text to indicate blue bars but have left references to the orange bars unchanged. Because there are no other bars that could be confused as either red or orange, we do not think this potential ambiguity will affect interpretability. We will continue to monitor this situation through the PDF conversion process during the final stages of submission.

---

## [Editor Report · Decision Letter 2]

26 Jan 2025

Multisensory stimuli facilitate low-level perceptual learning on a difficult global motion task in virtual reality

PONE-D-24-31083R2

Dear Dr. Huxlin,

We’re pleased to inform you that your manuscript has been judged scientifically suitable for publication and will be formally accepted for publication once it meets all outstanding technical requirements.

Kind regards,

Patrick Bruns

Academic Editor

PLOS ONE
---

## [Editor Report · Acceptance letter]

PONE-D-24-31083R2

PLOS ONE

Dear Dr. Huxlin,

I'm pleased to inform you that your manuscript has been deemed suitable for publication in PLOS ONE. Congratulations! Your manuscript is now being handed over to our production team.

Kind regards,

on behalf of

Dr. Patrick Bruns

Academic Editor

PLOS ONE